# DyStopia: Into a potential future of IEEE VIS under Plan S

Lonni Besançon [ID]*
Linköping University

Cody Dunne [ID]†
Northeastern University

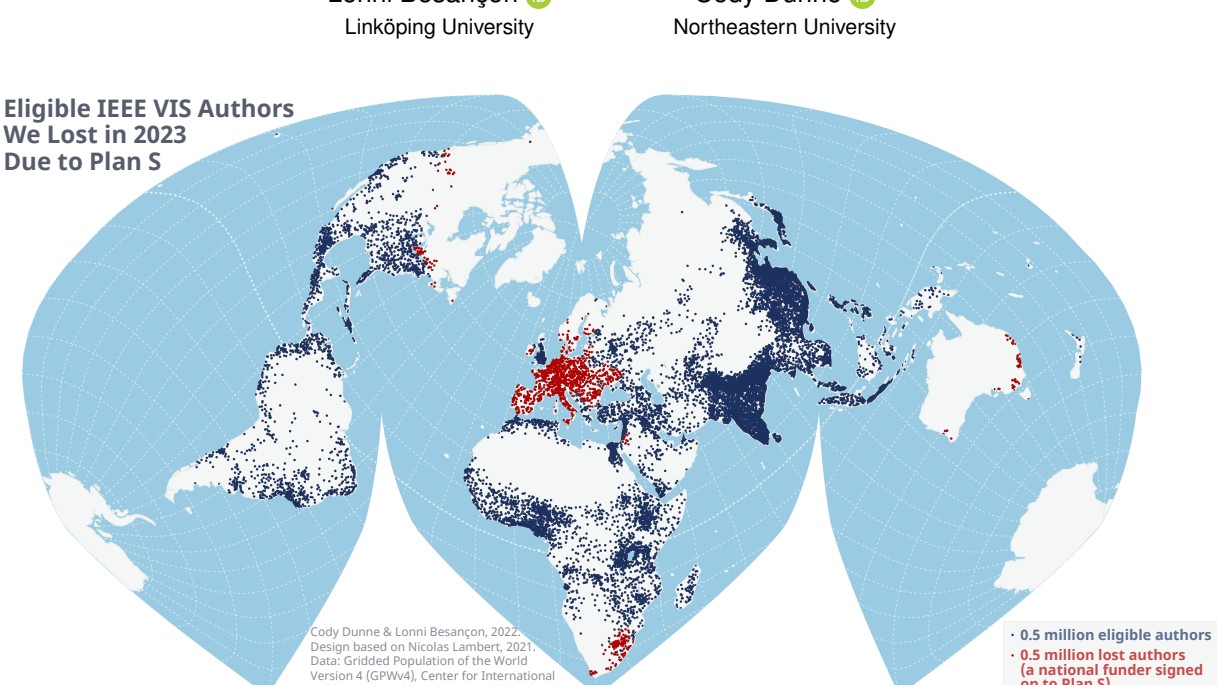

Figure 1: 2023 was the first year that IEEE VIS authors felt the full brunt of Plan S compliance. Many scientists in Europe and around the world discovered that their funding agencies prohibited them from publishing in IEEE TVCG because the journal was not truly open access. The lack of a clear plan from IEEE to address this predictable problem effectively excluded many authors from VIS, leading to the conference's substantial decline in submission and attendance in the following decades. *Note from 2022: lost authors was speculated.* Online at `https://observablehq.com/@codydunne/vis-plan-s`, archived at `https://osf.io/szvnq`.

## ABSTRACT

In this manuscript, we take a look at the dystopian future that could lie ahead of VIS should we fail to consider the impact of Plan S. It is the year 2030 and researchers from Europe have not been submitting their research to IEEE VIS for a number of years. We quickly retrace what led to this unfortunate outcome in the hope to avoid it and start an important conversation that would raise awareness in the VIS community about the importance and immediacy of Plan S.

## 1 SUPPLEMENTAL MATERIAL

A copy of this paper along with all supplemental materials is available at `https://osf.io/x75s6`. This includes the figures, code code used to generate them, data on the countries VIS first authors have hailed from between 1990 and 2019, and data on VIS registrations 2015–2021.

## 2 A RETROSPECTIVE UNTIL PLAN S

In 2018, cOAlition S, a group of European national funders with the help of Science Europe, the Scientific Council, and the European Research Council decided to adopt and sign Plan S [9]. In essence, Plan S is summarized by cOAlition S with the following:

---

*e-mail: lonni.besancon@gmail.com
†e-mail: c.dunne@northeastern.edu

"With effect from 2021, all scholarly publications on the results from research funded by public or private grants provided by national, regional and international research councils and funding bodies, must be published in Open Access Journals, on Open Access Platforms, or made immediately available through Open Access Repositories without embargo."[1]

The rationale behind such a strong push for Open Access was simple: **access to scientific knowledge should be universally possible regardless of a person's knowledge, funding, or affiliation**. As such, subscription models and paywalls were hindering access to a significant portion of scientific knowledge—despite much of this knowledge obtained thanks to publicly-funded academic ventures. Back in 2018, many researchers postulated that these monetization models only served the ever-growing business of science publishers and profoundly hindered access to knowledge and new advances.

The COVID-19 pandemic (c. 2020–2022) could have pushed even further the adoption of Open Access, which had been partially adopted across publishers to facilitate an international research effort [6]. Unfortunately, it seems that many publishers only used this opportunity to conduct an international Open Access washing and avoid backlash [2].

As the research presented at the IEEE VIS conference was not directly focused on COVID-19, the visualization community did not experience the potential benefits of full Open Access publications during COVID. Therefore, the conference organization did not make a strong push for Open Access to become the norm in the community. Consequently, when Plan S was put into motion in 2022 (after

---

[1]`https://www.coalition-s.org`

several years of forewarning), much of the VIS community was still unaware of what Plan S actually entailed. This lack of exposure to the looming issues has made planning and discussing within the community difficult.

## 3 A DYSTOPIAN FUTURE

While the ACM had drafted its plan for Plan S compliance in 2021, a year before the plan went into effect,[2] the IEEE had not come up with an effective plan in time. This led European researchers, as well as other teams around the world affected by Plan S requirements (see Fig. 1), to increasingly worry about whether they would be able to have their work presented at IEEE VIS. The 2022 statement from the U.S. White House declaring that American-state funded research must be Open Access [8] did not appear to accelerate IEEE's planning for Open Access publications—IEEE was also caught unprepared when the U.S. National Science Foundation in 2024 adopted requirements remarkably similar to Plan S. As IEEE had not started any negotiations or arrangements towards a full transition to Open Access within a reasonable timeframe,[3] authors from European countries gradually stopped submitting to IEEE VIS. In the first 29 years of VIS, 38.4% of the papers had first authors from a country with a national funder who signed on to Plan S[4]—a huge portion of the conference for IEEE to risk losing.

It is now the year 2030, and VIS consequently does not propose any research from European publicly-funded researchers, nor others who were also affected by Plan S requirements. In Fig. 2, we illustrate how the conference initially only lost a small portion of its attendees—but in short order suffered a rapid decline in attendance as more and more authors moved to other more open venues.

As the effect of Plan S had rightfully been anticipated to lead to inequitable outcomes on other countries and institutions than the ones directly involved with Plan S [4, 7], it is today regrettable to see our community divided and to imagine the wasted research opportunities that a unified community and venue would propose. This division of the visualization community has had a profound and negative impact on the ties between its members as much as on the produced research. The Plan-S-compliant community, not being able to attend VIS, has not substantially interacted with many of its former colleagues in over five years. While it is difficult to quantify the impact of such a split in the community, there is evidence of researchers unknowingly ignoring work from other sub-communities and thus sometimes wasting precious research resources (e.g. funding, time, participants in studies). This waste could have been avoided should our community have stayed united.

In addition, this division has led to tension between collaborators across communities who argue about where to publish their work such that it would have an impact in their community. Graduating PhD students have similarly found it difficult to find jobs in other countries, where researchers belong to a different sub-community from the one where they presented their dissertation work.

Finally, the European community has also wasted precious time and effort finding new funding sources and management to create a conference that would uphold the Plan S requirements. We can only deplore this addition to the already-existing waste in research efforts (see, e.g., [3]). In Fig. 1, we see the impact of Plan S and the lack of clear plan within the visualization and IEEE communities—effectively creating two worlds of visualization research. It is regrettable that we, visualization researchers, did not try our best to find a collective solution with IEEE that would work for us all. If only we could go back in time...

---

[2]https://authors.acm.org/open-access/plan-s-compliance
[3]See principle #8 of the Plan S Principles.
[4]Between 1990 and 2019, 3795 VIS papers had first authors from countries with national funders who signed on to Plan S. The dataset we used [5] has 9874 papers with known first-author locations and 2403 additional papers with unknown locations. We did not count the latter in our percentage.

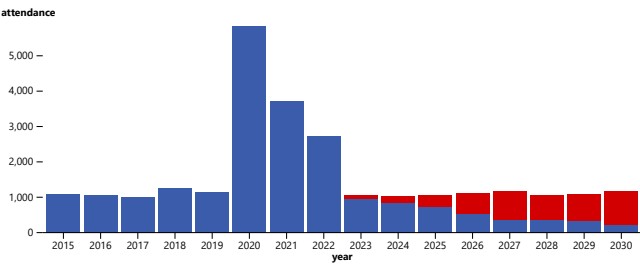

Figure 2: VIS registration records for 2015–2030 (**blue / bottom**) vs. predicted records if IEEE had planned ahead to provide Plan S compliance (**red / top**). (Note that during the years 2020–2023, VIS offered remote and then hybrid attendance options during the global COVID-19 pandemic, which led to temporarily-increased attendance.) We see an initial decline in registration as authors with Plan S funders submit elsewhere. However, within just a few years the rate of decline dramatically increases as researchers—unable to meet all their colleagues at VIS—flock to other conferences. *2022 note: Data from 2022 onwards is speculated.* Online at https://observablehq.com/@codydunne/vis-plan-s-attendance, archived at https://osf.io/7ydvq.

## 4 SNAP BACK TO REALITY OPE THERE GOES GRAVITY

While the above vision is purposefully dystopian and assuredly one of the worst possible outcomes of the current situation, it remains that the Plan S discussion and their conclusion are likely to impact our community at large through collaborations, hiring decisions, and challenges in finding or creating a new reliable venue. Yet, the situation is not *that* catastrophic. In May 2022 the IEEE reached an agreement with the UKRI (funding research body in the UK) that would allow researchers funded by the UKRI to deposit their Accepted Manuscripts in a repository at the time of final publication, with no embargo period, and under a CC-BY license.[5] Shortly before the acceptance of this manuscript, the U.S. White House also issued a statement that U.S. federally-funded research (and its artifacts) should be Open Access [8]. Unfortunately, and to the best of our knowledge, publishers (including IEEE) have not yet responded to this statement with any practical implications or solutions. This is in part due to the current ambiguity of the situation—the U.S. National Science Foundation has not announced the specifics for how it will be complying with these guidelines.

Papers accepted to the IEEE VIS conference appear in the IEEE Transaction on Visualization and Computer Graphics (IEEE TVCG). IEEE TVCG is a hybrid journal which means that most articles are not open access, and authors must pay a fee to have their articles available as Open Access and retain their copyright such that the paper is published under a CC-BY license. Until a transformative agreement is in place with cOAlition S, TVCG is not a viable solution for Plan S authors. The routes available to authors for Plan S compliance include several solutions, all of which would have different and difficult-to-predict consequences on the community.

### 4.1 Publishing in a journal outside of IEEE TVCG

This solution, the most drastic, is likely not feasible. The visualization community and IEEE have very strong ties and the VIS conference is inherently linked to the IEEE TVCG journal. Should Plan-S-compliant institutions opt for this solution, the community will likely still be divided and the newly-created journal would probably not have the impact of IEEE TVCG and could thus negatively impact the prospects of Early-Career Researchers. One of the advantages of this solution could be newly-acquired flexibility in how the journal operates in terms of publication format and deadlines (e.g.,

---

[5]https://open.ieee.org/ieee-compliance-with-ukri/

adopting a rolling deadline through the year like CSCW[6] or adopting Registered Reports for empirical visualization research [1]).

## 4.2 Publishing in a Plan-S-compliant journal from IEEE

IEEE has, in an effort to assist authors from Plan-S-compliant funders, created a compliant journal that accepted VIS papers could be published in. The journal, the IEEE Open Journal of the Computer Society, currently does not have near the reputation of IEEE TVCG and is thus also likely to negatively impact Early-Career Researchers. Moreover, this approach splits VIS research into two separate and inherently unequal journals.

## 4.3 Making IEEE TVCG Fully Open Access

As it stands, IEEE TVCG is a hybrid journal, but making it Fully Open Access would fulfill the requirements laid out by Plan S. However, this transition is likely to harm the revenues made by IEEE and alternative revenue sources would have to be found.

## 4.4 Creating transformative agreements

The possibility to create transformative agreements with Plan-S-compliant institutions is an interesting one. IEEE could reach an agreement with the institutions that—in exchange for a slightly higher subscription fee—their affiliated researchers could publish their work respecting the Plan S requirements. The IEEE as already reach such transformative agreements with some countries (e.g. Finland[7]) or institutions (e.g. CERN[8]), but this solution would only be temporary (per Plan S requirements[3]), and might still be ill-defined in the case of only one author belonging to a Plan-S-compliant institution. Indeed, in many cases, such transformative agreements only apply when the first author is from an institution covered by the transformative agreement. This could still be adopted while our community and IEEE figure out, together, a solution that is unlikely to leave anyone behind.

## 5 CONCLUSION

In this paper we have explored a very dystopian scenario of the future of IEEE VIS under Plan S. This scenario was purposefully dramatic and provoking, but we hope that it helped alert the community that Plan S is not just a European issue. Plan S covers authors in several other countries, but more importantly the effect of plan S will be felt by our entire community. Many less pessimistic scenarios could be envisioned and we have laid out some potential solutions to the issues. Other solutions could also be envisioned and we know that IEEE and many VIS committees are actively working on the issue.

## ACKNOWLEDGMENTS

This work was supported in part by the Knut and Alice Wallenberg Foundation (grant KAW 2019.0024).

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
