# OpenReview forum: "DyStopia: Into a potential future of IEEE VIS under Plan S"
_IEEE.org/2022/Workshop/altVIS — Accept_

### Official Review · Reviewer_xWzE · 2022-08-05

**Review:**

The authors make an important point (it is urgent that, as a community, we move towards Open Access), and I think that the format (looking back from an hypothetical future) falls under the banner of "alternative".  I therefore think this has the potential to be a good alt.vis paper.

However, the current article is very short, and I think it would be substantially improved by a more complete explanation of what the problem actually is.

I suspect that many in the community do not fully understand the problem, and it is precisely these people who most need to hear the message of this paper.

The initial part of Section 2 ("A retrospective from VIS 2030") covers events up to the present; I suggest splitting this into a separate section, and inserting additional explanation before describing the potential consequences of doing nothing.


## What readers need to know to understand the problem

I think that for the paper to be fully understandable in isolation, it would need to communicate most of the following:

Papers accepted to IEEE VIS appear in IEEE TVCG, which is a hybrid journal: most articles are not open access, and authors must pay a fee to have their articles available as Open Access.

There are three routes for Plan S compliance Section 2 of [1]:

* publish in a fully Open Access venue (which TVCG isn't - it's hybrid)
* publish in a hybrid subscription journal, but make the article Open Access and cover any Open Access fees with a Transformative agreement - i.e., a subscription contract where university libraries pay more in exchange for not having to pay OA fees (but the IEEE hasn't entered into any such transformative agreements)
* publish in a subscription journal, but publish the Author's Accepted Manuscript or the Version of Record in a repository (but this doesn't really work either, for reasons described below)

Publishing in a hybrid subscription journal but paying the OA fees isn't a solution, because many of the funder have refused to pay such fees.

In any case, "the publication must be openly available immediately with a Creative Commons Attribution license (CC BY) unless an exception has been agreed by the funder"

However, to apply a CC license to the Author Accepted Manuscript we would also need to opt-out of transferring copyright to the IEEE, and their policies say that this is not possible:

> Prior to publication by the IEEE, all authors or their employers shall transfer to the IEEE in writing any copyright they hold for their individual papers. Such transfer shall be a necessary requirement for publication, except for material in the public domain or which is reprinted with permission from a copyrighted publication.

(IEEE Policies, Section 6.3.1 - IEEE COPYRIGHT POLICIES, p. 76 of [2]).

Similarly, 8.1.9 D (p.89 of [2]) states:

> Upon submission of an article to IEEE, an author is required to transfer copyright in the article to IEEE, and the author must update any previously posted version of the article with a prominently displayed IEEE copyright notice (as shown in 8.1.9.B).

As this transfer is required *on submission*, the authors can't apply a CC license to the Author Accepted Manuscript, since they already transferred away copyright (when the manuscript is still the Author Submitted version).

There is apparently an opt-out for the new policy [2], but I think we need guidance on what we are expected to do when submitting to IEEE conferences and journals.


However, I'm told that some people have just been sticking Rights Retention Statements into the Acknowledgements section of papers submitted to IEEE journals, and have had them published without a problem [3].



This leaves a few potential solutions:

* switch to publishing in a journal other than TVCG (very difficult - IEEE effectively "own" the conference)
* make TVCG fully open-access (would cause the IEEE to lose revenue)
* creating Transformative Arrangement contracts covering TVCG with the IEEE (would require the IEEE to negotiate with many institutions, but other publishers have managed)
* allowing authors to retain copyright of the Author's Accepted Manuscript, so that they can release it immediately under a CC license

## The UKRI-IEEE agreement

In May the IEEE reached an agreement [4] with the UKRI (the umbrella funding government research funding council in the UK, which encompasses the subject-specific councils such as the Engineering and Physical Sciences Research Council).

This allows researchers funded by the UKRI to deposit their Accepted Manuscripts in a repository at the time of final publication, with no embargo period, and under a CC-BY license.

This means that UK researchers no longer face a looming compliance problem, so the map and statistic "36.8% of the papers had first authors from a country with a national funder" should be updated.

## Minor comments

* I'm not sure that the "into" in the title is necessary

* I think that the statement "the papers had first authors from a country with a national funder" is actually meant to be ""the papers had first authors from a country with a national funder *who signed on to Plan S*"

[1]: https://www.coalition-s.org/addendum-to-the-coalition-s-guidance-on-the-implementation-of-plan-s/principles-and-implementation/

[2]: https://pspb.ieee.org/images/files/files/opsmanual.pdf

[3]: https://www.coalition-s.org/blog/observing-the-success-so-far-of-the-rights-retention-strategy/

[4]: https://open.ieee.org/ieee-compliance-with-ukri/

**Conflicts:**

None.

**Review Inclusion:**

Yes

**Sufficiently Alt:**

Yes

**Superlative:**

Most pessimistic

---

### Official Review · Reviewer_6mn7 · 2022-08-24

**Review:**

The research is addressing the importance of open access publications in an innovative visualization of hypothetical scenarios. The article is lacking the details of discussions in related literature, methodology, and results.

**Conflicts:**

NA

**Review Inclusion:**

No

**Sufficiently Alt:**

Yes

**Superlative:**

Most wordless

---

### Official Review · Reviewer_zdBF · 2022-08-29

**Review:**

I like the topic, and I like the stand that the authors take. I like the approach and appreciate the urgency to raise awareness within our community. That being said, it looks like the paper was put together in a rush and does not fully exploit the potential of the idea/approach.

I think the authors could go one step further in their demonstraction/illustration of the potential future. For example, I would include a figure that shows the decline in submissions/participation to IEEE VIS/TVCG over the years, starting by plotting the historical data (number of submissions, number of papers, attendance, and proportion of these that are from countries under plan S) from the time it was collected, and then illustrate the (dystopian) decline from 2022 onwards.

They could also speculate on the dynamics of the research community, beyond "now there are two communities" but instead describe what negative impacts this could have. Some that come to mind are not citing across sub-communities, therefore ignoring existing work, reinventing the wheel, and costing millions of whatever currency one is using. It could lead to retaliation, such as the senior administrator in charge of hiring a new faculty vetoing the decision of the department to hire that person, because they are also editor in chief of the paywalled publishing platform, when this candidate has had to (or chose to) publish under plan S rules.

I would also like to read a bit more about potential solutions/approaches to solving this challenge. What are actionable steps the community could take? What are the tradeoffs of these different approaches? It would be nice to have the authors elaborate on 3 or 4 possible futures based on the decision the community makes, based on the path we choose to follow.

Minor comments:
- I think the teaser works well, however, instead of using a text explanation I would include a legend that indicates what a red dot is, and what a blue dot is.
- I'm not sure that "Note: this is speculation about one possible future." is needed. Given that the title makes it clear it is speculation, I would remove that sentence from the abstract that does not feel very "alt".

Typos/grammar:
- researchers from Europe have not been submitted -> have not been submitting
- what has happened to lead to -> what led to?
- As such, subscriptions models -> As such, subscription models
- thanks to publicly academic ventures

**Conflicts:**

I know and have worked (a bit) with the first author.
We have co-publised a book chapter.
One day, he slept on my sofa.

**Review Inclusion:**

Yes

**Sufficiently Alt:**

Yes

**Superlative:**

Most schismatic

---

### Official Review · Reviewer_ahyu · 2022-08-31

**Review:**

This paper is an apt and timely consideration of the effects of not embracing open access. It clearly fits the bill of being alt and being valuable to our community, and as such should be accepted. I advise the authors to read the careful and thorough reviews provided and, while time is short, embrace as many of the suggestions as possible. To these I would add several comments:
- Within the fiction I wondered: why would this not just lead to another conference that was plan s-compliant (e.g. SIGVIS)? What are the consequences of this split if any?
- Outside of it I wondered how the authors respond to the recent declaration that American state funded research must be open access? Perhaps it could be commented on in the paper.

**Conflicts:**

None

**Review Inclusion:**

Yes

**Sufficiently Alt:**

Yes

**Superlative:**

Most Speculative

---

### Decision · Program_Chairs · 2022-08-31

Accept